# Nitric Oxide Production and Effects in Group B *Streptococcus* Chorioamnionitis

**DOI:** 10.3390/pathogens11101115

**Published:** 2022-09-28

**Authors:** Mary Frances Keith, Kathyayini Parlakoti Gopalakrishna, Venkata Hemanjani Bhavana, Gideon Hayden Hillebrand, Jordan Lynn Elder, Christina Joann Megli, Yoel Sadovsky, Thomas Alexander Hooven

**Affiliations:** 1Department of Pediatrics, Vanderbilt University School of Medicine, Nashville, TN 37232, USA; 2Department of Pediatrics, University of Pittsburgh School of Medicine, Pittsburgh, PA 15224, USA; 3Manual Hematology and Coagulation Department, The Cleveland Clinic, Cleveland, OH 44195, USA; 4Department of Obstetrics, Gynecology, and Reproductive Sciences, University of Pittsburgh School of Medicine, Pittsburgh, PA 15213, USA; 5UPMC Magee-Womens Research Institute, Pittsburgh, PA 15213, USA; 6UPMC Children’s Hospital of Pittsburgh Richard King Mellon Institute for Pediatric Research, Pittsburgh, PA 15224, USA; 7UPMC Children’s Hospital of Pittsburgh, 4401 Penn Ave. Rangos Research Building #8128, Pittsburgh, PA 15224, USA

**Keywords:** group B *Streptococcus*, *Streptococcus agalactiae*, chorioamnionitis, innate immunity, nitric oxide, transcriptomics

## Abstract

Intrauterine infection, or chorioamnionitis, due to group B *Streptococcus* (GBS) is a common cause of miscarriage and preterm birth. To cause chorioamnionitis, GBS must bypass maternal-fetal innate immune defenses including nitric oxide (NO), a microbicidal gas produced by nitric oxide synthases (NOS). This study examined placental NO production and its role in host-pathogen interactions in GBS chorioamnionitis. In a murine model of ascending GBS chorioamnionitis, placental NOS isoform expression quantified by RT-qPCR revealed a four-fold expression increase in inducible NOS, no significant change in expression of endothelial NOS, and decreased expression of neuronal NOS. These NOS expression results were recapitulated ex vivo in freshly collected human placental samples that were co-incubated with GBS. Immunohistochemistry of wild type C57BL/6 murine placentas with GBS chorioamnionitis demonstrated diffuse inducible NOS expression with high-expression foci in the junctional zone and areas of abscess. Pregnancy outcomes between wild type and inducible NOS-deficient mice did not differ significantly although wild type dams had a trend toward more frequent preterm delivery. We also identified possible molecular mechanisms that GBS uses to survive in a NO-rich environment. In vitro exposure of GBS to NO resulted in dose-dependent growth inhibition that varied by serovar. RNA-seq on two GBS strains with distinct NO resistance phenotypes revealed that both GBS strains shared several detoxification pathways that were differentially expressed during NO exposure. These results demonstrate that the placental immune response to GBS chorioamnionitis includes induced NO production and indicate that GBS activates conserved stress pathways in response to NO exposure.

## 1. Introduction

*Streptococcus agalactiae* (group B *Streptococcus*; GBS) remains a leading cause of global perinatal morbidity and mortality [1,2,3,4,5]. GBS is a common commensal organism of the genitourinary tract, where it rarely causes symptoms in nonpregnant adults [6,7,8,9,10,11,12]. However, in colonized pregnant women, GBS can vertically ascend from the vaginal tract to the fetal membranes, amniotic fluid, and placenta causing chorioamnionitis prior to delivery [13,14,15]. Chorioamnionitis increases the risk of miscarriage, stillbirth, and preterm birth [16,17,18]. Because chorioamnionitis often precedes labor, its impact is not significantly affected by recommendations for intrapartum antibiotic prophylaxis in pregnant women colonized with GBS [19,20], leaving a large population of maternal-fetal dyads at risk for invasive disease and poor outcomes.

In order to survive in the pregnant uterus and cause chorioamnionitis, GBS must circumvent host immunity. At the level of the placenta, there is a complex interplay of immune cells, trophoblasts, and defense molecules that serve to protect the fetus from invading pathogens while maintaining immune tolerance to fetal antigens [21]. One key immune defense molecule is nitric oxide (NO), which is produced in the placenta at baseline and increases in concentration during bacterial invasion [22]. Despite evidence that NO is present in the placenta, little has been published about its role in host-pathogen interactions during chorioamnionitis.

NO is formed from L-arginine by the enzyme nitric oxide synthase (NOS), of which there are three isoforms: endothelial, neuronal and inducible. The NOS isoform produced by trophoblast cells of the human placenta is predominantly endothelial (eNOS) as it is essential for blood vessel smooth muscle relaxation, and aberrant production has been implicated in preeclampsia [23,24]. Neuronal NOS (nNOS) has also been localized to syncytiotrophoblast cells, though the role of nNOS in the placenta is unclear [25]. Unlike the endothelial and neuronal NOS isoforms, which are thought to be constitutively expressed, inducible NOS (iNOS) shows increased expression under certain conditions such as infection [26]. iNOS has been localized to fibroblasts from human chorion and amnion as well as decidual macrophages in term pregnancies [25]. During infection, NO generated by iNOS is responsible for vasodilation, induction of cytokine signaling, and conversion to reactive oxygen species and reactive nitrogen intermediates that lead to bacterial cell death [25,26].

Bacteria differ in their abilities to withstand NO exposure. Multiple human pathogenic bacteria, which face elevated NO concentrations at infection sites, encode stress response genes that are activated by NO-mediated oxidative damage, promoting bacterial survival and further invasion [25,26]. More broadly, bacteria have been shown to employ mechanisms to repair damage caused by reactive nitrogen species, detoxify the environment, and inhibit NOS [27]. While the bactericidal effects of NO are initially adaptive, overproduction of NO can affect host tissue viability as a result of blood flow dysregulation and direct oxidative damage. In the placenta, high concentrations of NO and its metabolites in the setting of chorioamnionitis are thought to contribute to placental cell apoptosis, potentially leading to preterm labor or fetal demise [28].

GBS invades and causes chorioamnionitis despite a robust placental immune response that includes NO production. GBS tolerance of host stressors is enhanced by virulence factors associated with increased adherence and penetration of vaginal epithelium, intrauterine infection, and preterm birth [29,30,31,32,33,34,35,36]. One well-described GBS virulence factor is β-hemolysin/cytolysin, a pigmented, pore-forming toxin that leads to eukaryotic cell death [37,38,39,40,41,42,43]. However, no GBS virulence factors have been identified as specifically enabling GBS to withstand NO exposure in the placenta.

In this study we use a murine model of ascending GBS chorioamnionitis and primary human placental samples in combination with unbiased, bacterial genetic tools to examine the role of NO on GBS-related adverse pregnancy outcomes and to examine GBS molecular pathways that promote evasion of placental NO.

## 2. Methods

### 2.1. Ethics Statement

Animal experiments were conducted under an approved IACUC protocol (protocol #20016575) at University of Pittsburgh. Adult phlebotomy for hemolysis assays and collection of human placental samples were performed under University of Pittsburgh approved IRB protocols (protocols #19110106 and #19100322, respectively). All subjects provided informed consent for blood or tissue collection after disclosure of relevant risks and benefits.

### 2.2. Statistical Analyses

Statistics for RNA-seq were calculated with DESeq2 (Bioconductor, Buffalo, NY), which uses the Benjamini-Hochberg adjustment for multiple comparisons. Gene expression differences were considered statistically significant if they were at least twofold greater or less than the control condition using a false discovery rate of 10%. Statistics for remaining figures were calculated using Prism v 9.0.0 for macOS (Graphpad Software, San Diego, CA, USA). A one-sample t and Wilcoxon test was used for analysis of NOS isoform gene expression quantification. Preterm delivery rates among WT and iNOS- mice were analyzed with the log-rank test. Rates of intrauterine fetal demise (IUFD) in WT and iNOS- mice were analyzed with chi-square tests. The hemolysis assay was analyzed with a one-way ANOVA with Bonferroni’s correction for multiple comparisons. Comparison of survival between GBS strains CNCTC 10/84 and 10/84 Δ*cylE* was performed via an unpaired t test. RT-qPCR validation of DETA-responsive genes used ANOVA with Dunnett’s correction for multiple comparisons.

### 2.3. Bacterial Strains and Growth Conditions

The following GBS strains were utilized: CNCTC 10/84 (serotype V, sequence type 26), 10/84 Δ*cylE*, A909 (serotype Ia, sequence type 7), and CDC Active Bacterial Core Surveillance *Streptococcus agalactiae* strains 2008232728 (serotype Ia), 2008232729 (serotype Ib), 2008232738 (serotype II), 2008232582 (serotype III), 2011201884 (serotype IV), 2008232731 (serotype V), 2010228816 (serotype VI), 4832-06 (serotype VII), 5030-08 (serotype VIII), and 7509-07 (serotype IX) [44,45]. All GBS strains were grown in tryptic soy (TS) medium (Fisher Scientific, cat. # 211822) at 37 °C under stationary conditions. No antibiotic selection was required for any GBS strains utilized in these experiments. The GBS strain 10/84 Δ*cylE* was created by two step allelic exchange as previously described [46].

### 2.4. Murine GBS Chorioamnionitis Model

Mouse strains C57BL/6J and B6.129P2-*Nos2^tm1Lau^*/J (The Jackson Laboratory, cat # 000664 and 002596) were used for the ascending GBS chorioamnionitis model. The B6.129P2-*Nos2^tmLau^*/J mice, also denoted iNOS-, are homozygous for the targeted *Nos2^tmLau^* mutation by which they are unable to produce iNOS, and, during strain development, the mice were backcrossed into a C57BL/6J background [47].

Chorioamnionitis was induced with adaptations from a previously described model (see Figure 1) [41]. Timed pregnancies were established in wild type (WT) or iNOS- mice and pregnancy was confirmed by examination and consistent weight gain over the first 12 days of pregnancy. On gestation day 13, GBS strain CNCTC 10/84, grown to stationary phase, was pelleted and resuspended in PBS in a 1:1 mixture with 10% sterile gelatin to increase viscosity. The GBS mixture was diluted and plated to determine the intravaginal inoculum, which corresponded to ~10^8^ CFU/mL. Dams were then intravaginally inoculated under isoflurane anesthesia with 50 µL of the GBS mixture. Mice were subsequently single-housed and monitored daily for general wellbeing, weight gain or loss, and for preterm delivery. If the dams had not delivered prematurely by gestation day 17 they were euthanized and dissected to evaluate for colonization and infection.

Vaginal swabs were obtained immediately following sacrifice on gestation day 17 to determine the presence of GBS vaginal colonization. A sterile nasopharyngeal swab was moistened with 250 µL sterile PBS, inserted into the murine vagina and rotated 5 times then placed into 500 µL sterile PBS to release adherent GBS. The PBS was then serially diluted and plated on GBS-specific CHROMagar plates (DRG International, cat # SB282) to determine CFU/mL. 

All murine placentas and fetuses were dissected from each dam’s uterus and processed for RNA extraction (placenta and pup), fixation (placenta only), or snap freezing (placenta only). IUFD was determined based on visual inspection for hemorrhagic or purulent necrosis or anatomical resorption. Placentas used for RNA extraction were cut into halves and the torso and abdomen of fetuses were separated. One half of the placenta and the fetus torso was homogenized in 500 µL sterile PBS using a bullet blender (Next Advance, model BBX24), serially diluted, and plated on GBS-specific CHROMagar plates to determine the presence of GBS chorioamnionitis and to enumerate CFU/mL of the tissue homogenate. The other half of the placenta underwent manual RNA extraction following homogenization (Cole-Parmer, PRO Scientific Bio-Gen PRO200 Homogenizer, item # UX-04751-25) using the MagMAX *mir*Vana Total RNA Isolation Kit (Applied Biosystems, cat # A27828), according to manufacturer instructions. Other placentas were fixed by incubation of whole placenta in 4% paraformaldehyde for 48 h, after which paraformaldehyde was replaced with 70% ethanol and stored at room temperature or snap frozen in 100% ethanol with dry ice and placed at −80 °C for storage.

### 2.5. Human Placental Collection

Placentas were collected through the Steve C. Caritis Maternal Obstetric Biobank at the University of Pittsburgh. Inclusion criteria included cesarean delivery at greater than 35 weeks of pregnancy and singleton gestation. Patients were excluded if they were in labor, had preterm or prolonged (>18 h) rupture of the fetal membranes, infection, or underlying inflammatory conditions. Placentas were harvested immediately after delivery and a placental fragment was placed into DMEM F12 media. Chorionic villi were isolated and dissected into 0.5 × 0.5 cm segments in PBS and rinsed in DMEM. These were placed in DMEM F12 with 10% FBS. Chorionic villi explants were incubated for 1 h at 37 °C with 5% CO_2_ prior to infection.

### 2.6. Human Placental GBS Infection and RNA Purification

Bacterial cultures were grown overnight and diluted to OD_600_ = 0.5. Bacteria were centrifuged and resuspended in PBS to a density of 10^6^ CFU/mL, which was confirmed by plating serial dilutions for colony enumeration. A total of 10 μL of the bacterial suspension was used to inoculate chorionic villi explants. Serial dilutions were performed of the inoculum and bacteria were plated on blood agar. Three hours post-infection, the tissue was harvested and lysed in RNA lysis buffer. RNA was purified using the GenElute Mammalian Total RNA Miniprep kit and its associated DNAse Digestion Kit (Sigma, RTN 350) according to the manufacturer instructions.

### 2.7. Quantification of Placental Gene Expression

NOS isoform gene expression was quantified from placentas by RT-qPCR. RNA from experimental and control placenta samples was extracted as described above. qPCR was performed using TaqMan Fast Virus 1-Step Master Mix (Thermo, cat # 4444432) and run on a CFX96 Touch Real-Time PCR Detection System (BioRad) using the paired CFX Manager 3.1 software. A PrimeTime qPCR probe assay for Polr2a was used as the mouse housekeeping gene (Mm.Pt.39a.22214849, Integrated DNA Technologies, Coralville, IA, USA). Three additional PrimeTime Mini qPCR assays for mouse iNOS, eNOS, and nNOS were used to quantify NOS isoform expression (Mm.PT.58.10244018, Mm.PT.58.30784006, and Mm.PT.58.11089624, Integrated DNA Technologies). Corresponding human NOS isoform gene PrimeTime Mini qPCR assays were used to quantify iNOS, eNOS, and nNOS (Hs.PT.58.14740388, HS.PT.58.21447620, Hs.PT.58.24422037, Integrated DNA Technologies). We used a YWHAZ expression assay (Hs.PT.39a.22214858, Integrated DNA Technologies) as the human placental housekeeping gene for transcript normalization [48].

### 2.8. Histological Preparation and Staining of Murine Placenta

Whole placentas that were paraformaldehyde fixed, as described above, were embedded in paraffin and sectioned onto glass slides. Hematoxylin and eosin staining and periodic acid-Schiff (PAS) staining were performed using standard protocols. Slides for immunohistochemistry were exposed to rabbit anti-mouse iNOS antibody at a 1:100 concentration (Abcam, cat # ab15323) and secondary biotinylated goat anti-rabbit IgG antibody at 1:200 (Vectorlab, cat # BA-1000), then stained with the Vectastain ABC-HRP kit (Vectorlab, cat # PK-4000) according to manufacturer recommendations. 

### 2.9. Growth Curve Analysis

GBS strains were grown overnight to late-log phase in TS broth then all cultures were normalized to OD_600_ = 1.0 the following morning. Normalized GBS cultures were added individually as a 1:50 dilution to TS broth with graded concentrations of diethylenetriamine/nitric oxide adduct (DETA NONOate, NOC-18, Santa Cruz Biotechnology, cat # sc-202247B). A stock solution of 100mM of DETA NONOate was prepared, filter sterilized, and then serially diluted to 20 mM, 10 mM, 5 mM, 2.5 mM, and 1.25 mM in TS, respectively. DETA NONOate is a diazeniumdiolate that releases nitric oxide into aqueous solution at a steady rate with a half-life of approximately 20 h [49,50]. Samples were loaded into a 96-well clear, flat bottom plate and covered with a lid that had been pre-treated with sterile defogging solution. The absorbance (OD_600_) was read every 10 min for 18 h at 37 °C using a Molecular Devices SpectraMax M4 (San Jose, CA, USA). 

### 2.10. Hemolysis Assays

Hemolysis assays were performed with whole-cell GBS on washed human erythrocytes as previously reported with modifications to bacteria preparation [51]. GBS strain CNCTC 10/84 was grown to late-log phase, normalized to OD_600_ = 1.0 and then added as a 1:50 dilution to 5 mM, 2.5 mM, and 1.25 mM DETA NONOate in TS broth. The bacterial cultures were then grown overnight in the DETA NONOate-TS mixture at 37 °C. The following day the GBS cultures were spun and washed with PBS to remove any remaining DETA NONOate in solution prior to performing the hemolysis assays.

### 2.11. GBS Survival Assay

GBS strains CNCTC 10/84 and 10/84 Δ*cylE* were grown individually from single colonies in TS broth to an OD_600_ = 1.0. The cultures were then pelleted and washed with sterile PBS, after which they were re-pelleted and resuspended in sterile PBS to an OD_600_ = 1.0. A stock solution of 20 mM DETA NONOate was created using an aliquot of normalized GBS in PBS then diluted to 10 mM in a second aliquot of normalized GBS in PBS. The cultures were incubated in a 37 °C water bath for two hours. A sample of each culture was plated and serially diluted to enumerate CFU/mL at the start of the outgrowth and 2 h later, at the end of the outgrowth. 

### 2.12. Whole-Genome RNA-Seq Transcriptomic Analyses

Duplicate samples of GBS strains CNCTC 10/84 and A909 were grown overnight from single colonies to late-log phase in TS broth at 37 °C then normalized to OD_600_ = 1.0. A 1:50 dilution of each bacterial strain was added to 13 mL of 1.5 mM DETA NONOate in TS broth and grown until OD_600_ reached 1.0. The GBS was then pelleted and RNA extracted using the RiboPure RNA Purification Kit (Invitrogen, cat # AM 1925) and DNase treated per manufacturer instructions using provided reagents. 

The Illumina TruSeq total RNA kit was used for preparation of the sequencing library, and the Ribo-Zero Plus rRNA Depletion Illumina kit was used for rRNA depletion. Random primers were used to initiate first and second strand cDNA synthesis. The 3′ ends were adenylated then adapters ligated and the sequencing library was then amplified with indexing. Sequencing was performed on an Illumina NextSeq500 platform over 150 cycles with paired-end 75-nt reads. 

Reads were trimmed, demultiplexed, and aligned to the reference 10/84 and A909 genomes using Bowtie 2 (Johns Hopkins University, Baltimore, MD, USA) [52]. DESeq2 (Bioconductor, Buffalo, NY) was used to calculate statistical analysis of the alignments and provided the log fold change in gene expression with corresponding Benjamini-Hochberg adjusted *p*-values [53]. Genome2D was used for gene set enrichment analysis [54]. Orthologous genes were determined by comparative strain analysis using the Microbial Genome Database for Comparative Analysis (MBGD) [55,56].

### 2.13. RT-qPCR Validation of RNA-Seq Up- and Downregulated Genes

Whole RNA isolated from DETA NONOate or vehicle control exposed GBS was purified and treated with DNase as described above. cDNA was generated from the RNA samples using the Applied Biosystems High-Capacity cDNA Reverse Transcription Kit (Thermo cat # 4368814), with reverse transcriptase-negative samples prepared as negative controls according to manufacturer recommendations.

RT-qPCR was performed with Applied Biosystems Sybr Green PCR Master Mix (Thermo cat # 4309155) on a CFX96 Touch Real-Time PCR Detection System (BioRad) using the paired CFX Manager 3.1 software. Primers for the candidate genes validated by RT-qPCR, and the housekeeping gene *recA*—which was used as a normalization control—were as follows (gene/forward primer/reverse primer; primers in 5′-to-3′ orientation). *acetcoA*/TGG TTG AGT TTA ATG GTC TAG GG/CAC GGA TAG CAC GAA TTC TAG T. *cylE*/CAC GGA TAG CAC GAA TTC TAG T/GCT TTG CCA GGA GGA GAA TA. *fetB*/GCT TTG CCA GGA GGA GAA TA/GTC CCT GAC CCA ATA GCT ATA AA. *ribD*/CTA CAA AGA CGG GCG ATT CTA A/GAT ACC GAC CAT AAT AGC ACT ACA T. *recA*/GTG GGA TTG CTG CCT TTA TTG/CTG AGT CAG GTT GAG ACA AGA G.

## 3. Results

### 3.1. Placental iNOS Isoform Expression Is Increased in a Murine Model of GBS Chorioamnionitis

Using a previously described murine model of ascending GBS chorioamnionitis [41] in which mice are intravaginally inoculated with GBS strain CNCTC 10/84 or sham control (Figure 1), we characterized the placental NOS isoform expression profile during experimentally induced chorioamnionitis in WT C57BL/6J dams. Chorioamnionitis was confirmed, and bacterial load in the placenta quantified by the number of GBS colonies recovered on GBS-specific CHROMagar plates from samples of each placenta. The expression of inducible, endothelial, and neuronal NOS isoforms was quantified by RT-qPCR from placentas obtained from mice with GBS chorioamnionitis and controls. Samples from mice that did not develop colony-confirmed chorioamnionitis were excluded. A total of seven placentas from three chorioamnionitis mice were analyzed alongside four sham infected placentas from four mice. RNA from each placenta was assayed by RT-qPCR in three technical replicates. The expression of iNOS was increased four-fold during GBS chorioamnionitis (median fold-change = 4.07 by one sample t and Wilcoxon test), whereas eNOS expression was unchanged compared to sham-inoculated placentas (median fold-change = 0.85, Figure 2A). nNOS could not be detected in five murine placental samples (one sham and four choriomnionitis placentas). Among the three samples for which nNOS was detectable, its expression was decreased in WT mice with GBS chorioamnionitis (median fold-change = 0.19) relative to sham infected animals.

### 3.2. Ex Vivo Human Placental Samples Infected with GBS Show Increased iNOS Expression

Through an existing pregnancy biological sample repository program at the University of Pittsburgh, we collected late-preterm or term human placental samples following cesarean delivery in the absence of known infectious or inflammatory complications. Exclusion criteria included preterm or prolonged rupture of the fetal membranes, active labor, or clinical concerns for chorioamnionitis or other maternal infection. Immediately following sterile collection, the samples were dissected to isolate chorionic villi, which were then co-incubated with GBS strain CNTC 10/84. Whole RNA from GBS-exposed samples and unexposed matched controls were used for qPCR determination of NOS isoform expression. Two biological replicates were assessed in two separate qPCR experiments, each with triplicate technical replicates.

As in the murine model, human samples demonstrated statistically significant upregulation of iNOS expression following GBS infection. eNOS levels did not differ significantly across treatment condition. Unlike in the murine samples, nNOS transcript was not detected in the human placenta regardless of treatment condition (Figure 2B).

### 3.3. iNOS Isoform Expression Localizes to the Junctional Zone of GBS-Infected Murine Placentas

We performed immunohistochemistry to localize iNOS expression in the GBS-infected murine placenta. Samples with chorioamnionitis demonstrated diffuse iNOS isoform expression with increased and clustered signal intensity in the junctional zone, identifiable as a strongly pink staining band on PAS stain, as well as areas of abscess in the labyrinth (Figure 3). The sham-inoculated placentas, however, showed decreased iNOS staining overall and no discernable anatomic areas of heightened expression.

### 3.4. Wild Type Dams with GBS Chorioamnionitis Have Increased Rates of Preterm Delivery

We also sought to determine the effects of NO on maternal, pregnancy, and fetal outcomes by inducing GBS chorioamnionitis in an iNOS knockout (iNOS-) mouse strain, B6.129P2-*NOS2^tm1Lau^*/J generated in the C57BL/6 background. The iNOS- mouse strain is fertile, though they have been noted to have smaller litter sizes compared to WT mice [57]. Following intravaginal inoculation with GBS strain CNCTC 10/84, 70% of WT mice developed GBS chorioamnionitis compared to 80% of iNOS- mice (Figure 4A). Despite the fact that more iNOS- mice developed GBS chorioamnionitis, the WT mice showed a trend toward more preterm delivery, although the difference was not statistically significant (*p* = 0.14 by log-rank test, Figure 4B). The rates of IUFD between WT and iNOS- strains was similar at 32% and 35% of fetuses, respectively. Both rates were significantly increased compared to sham-inoculated controls (*p* < 0.0001 by chi-square, Figure 4C).

### 3.5. NO Exposure Results in Dose-Dependent GBS Growth Inhibition That Varies by Serovar

To determine the response of GBS to NO, we exposed 10 GBS strains from the CDC Active Bacterial Core Surveillance Program strain collection, representing all capsular serotypes as well as strain CNCTC 10/84, a capsular serotype V strain, to diethylenetriamine/nitric oxide adduct (DETA NONOate), a NO adduct that releases nitric oxide into liquid solution at a steady rate. The GBS strains were grown for 18 h in liquid culture in graded concentrations from 1.25 mM to 20 mM DETA NONOate and absorbance was measured every 10 min. We noted significant differences in growth kinetics among the strains (Figure 5A). GBS strain CNCTC 10/84 had better growth at all DETA NONOate concentrations, and was the only strain to grow at 10 mM DETA NONOate. In contrast, the serotype Ia GBS strain did not grow in the presence of DETA NONOate until approximately 16.5 h (Figure 5B,C), with other strains showing intermediate growth phenotypes.

### 3.6. β-Hemolysin/Cytolysin Confers a Growth and Survival Advantage to GBS in a NO-Rich Environment

β-hemolysin/cytolysin (βH/C) is a well-described virulence factor that enhances GBS survival and increases host tissue damage during infection [42]. βH/C has been shown to improve intracellular survival in innate immune cells [58], cause pore formation in numerous human cell types [38,41,59,60,61,62], activate inflammatory cytokine signaling [39,60], and provoke apoptosis [42]. βH/C is believed to be a non-protein ornithine-rhamnopolyene, a pigmented variant of the carotenoid family of molecules with antioxidant properties that may also increase GBS fitness during infection [63,64]. CNCTC 10/84, which demonstrated unusual NO resistance among the panel of tested strains, is hyper-hemolytic and hyperpigmented due to elevated βH/C expression resulting from under-expression of the two-component CovR/CovS system [65]. This observation raised the question of whether βH/C might contribute to GBS survival in NO-rich environments.

Hemolysis assays with CNCTC 10/84 and washed human erythrocytes, performed in graded concentrations of DETA NONOate, revealed that this GBS strain was still able to hemolyze human red blood cells in the presence of NO, though the degree of hemolysis decreased with increasing concentrations of DETA NONOate (one-way ANOVA with Bonferroni’s multiple comparisons test, Figure 6A).

To further elucidate the role of β-hemolysin/cytolysin in resisting the microbicidal effects of NO, we compared the growth and survival of WT CNCTC 10/84 with an isogenic, non-hemolytic Δ*cylE* strain incapable of producing βH/C. WT CNCTC 10/84 had delayed growth at all DETA NONOate concentrations but the growth delay was less than that of 10/84 Δ*cylE* (Figure 6B). This was most apparent at 10 mM and 5 mM DETA NONOate, at which time to mid-log phase between the two strains was significantly different (*p* < 0.0001 and *p* < 0.05 by one-way ANOVA with multiple comparisons). We then compared survival of WT CNCTC 10/84 and 10/84 Δ*cylE* in 10 mM DETA NONOate, as that concentration of DETA NONOate appeared to be the threshold that growth differences were most divergent between the two strains. We exposed each strain individually to 10 mM DETA NONOate in liquid culture then plated serial dilutions of each sample on TS agar following 2 h of NO exposure. The 10/84 Δ*cylE* strain had only 34% survival in the presence of DETA NONOate compared to the wild type strain (*p* < 0.01 by unpaired t test, Figure 6C), suggesting that βH/C contributes to GBS survival in conditions of NO exposure. 

### 3.7. GBS Displays a Broad Transcriptomic Response to NO Exposure

We suspected that GBS employs genetically regulated adaptive responses to environmental NO. To determine transcriptional changes due to NO exposure, we extracted and sequenced RNA (RNA-seq) from GBS strains CNCTC 10/84 and A909, a capsular serotype Ia comparator, following exposure to 1.5 mM DETA NONOate or vehicle only control growth. This sub-lethal DETA NONOate concentration was chosen to permit comparison between the more sensitive A909 strain and the more tolerant CNCTC 10/84 strain (Figure 5A). CNCTC 10/84 and A909 represent commonly isolated capsular serotypes in clinical neonatal infection [66]. RNA-seq revealed that both strains had genome-wide responses to NO and that there were distinct transcriptomic profiles following NO exposure (Figure 7, Appendix A). Figure 7A shows clusters of orthologous genes (COG) predicted functional classes of all shared genes with at least twofold upregulation or downregulation in response to DETA exposure. There were 142 shared upregulated and 179 shared downregulated genes between CNCTC and A909, while 772 had no change (Figure 7B). Individual genes with shared upregulation or downregulation greater than fourfold are listed in Table 1 and Table 2 along with consensus functional annotations determined with the PANTHER database [67]. Appendix A shows all orthologous genes with differential responses to DETA exposure, grouped by fold-expression thresholds. The most upregulated gene in response to DETA, in both CNCTC 10/84 and A909, was a bifunctional acetaldehyde-CoA/alcohol dehydrogenase (A909 gene locus SAK_RS00425 and CNCTC 10/84 gene locus W903_RS00440) (Table 1). There were 13 shared genes with at least a four-fold decrease in expression (Table 2), of which six were classified as ABC transporters in their NCBI-published genome annotations. *FetB*, a YBBM-related iron export transporter (A909 gene locus SAK_RS04090 and CNCTC 10/84 gene locus W903_RS03770) was the most downregulated gene, with a 16-fold decrease in gene expression following NO exposure. The *cyl* operon that influences βH/C production was slightly downregulated in both CNCTC 10/84 and A909 (LFC −1.97 and −0.91, respectively).

To validate our RNA-seq screen, we used RT-qPCR with primers matched to select up- and downregulated genes. Assessment of the *fetB* iron export transporter and the *cylE* gene demonstrated strong and modest downregulation, while the bifunctional acetaldehyde CoA/alcohol dehydrogenase and riboflavin biosynthesis (*ribD*) genes showed upregulation in the presence of DETA. These RT-qPCR validation assays were consistent with the genome-wide screen (Figure 7C).

## 4. Discussion

GBS is estimated to cause 57,000 stillbirths and several million preterm births worldwide each year [68]. GBS ascending infection during pregnancy, known as chorioamnionitis, is associated with stillbirth and preterm labor—much of which is unquantified in global surveys. While some of the host-pathogen interactions that underlie GBS chorioamnionitis have been elucidated, there is still a paucity of information about mechanisms that GBS uses to evade the placental immune system. 

This study explored the placental immune response of NO production, via the enzyme NOS, and its role in host-pathogen interactions in GBS chorioamnionitis. We determined that all three NOS isoforms—inducible, endothelial, and neuronal—are expressed by the murine placenta at baseline and during GBS chorioamnionitis. In an ex vivo chorioamnionitis model using primary human placental tissue, however, only eNOS and iNOS were detectable by RT-qPCR. In both murine and human samples, placental iNOS expression was significantly increased following GBS infection compared to sham-infected placentas. This indicates that the placenta utilizes iNOS expression and, therefore, NO production as a defense mechanism against GBS. eNOS isoform expression was unchanged, which is likely explained by the fact that eNOS is essential for maintaining vascularity of the placenta, and thus is essential for pregnancy viability. Alteration of eNOS expression has been shown to affect the integrity of the placenta leading to maternal disease and preterm birth [69]. This background and our qPCR expression data indicate that the murine pregnancy outcomes we observed were not confounded by changes in eNOS expression. In both murine and human samples, placental nNOS was minimally detected, if at all. In human placental tissue, nNOS expression was not detected at baseline or following GBS exposure. Among murine placentas infected with GBS for which nNOS was detectable (n = 3), its level was significantly decreased relative to sham infected controls. One possible explanation is that nNOS expression was downregulated so that L-arginine, the substrate for nitric oxide synthases, could be diverted toward iNOS utilization.

Once we confirmed placental iNOS expression during GBS chorioamnionitis, we performed immunohistochemistry on placentas to localize this enzyme. The placentas of sham-inoculated mice displayed only faint and diffuse iNOS antibody staining. The placentas from mice with GBS chorioamnionitis displayed widespread iNOS antibody staining with especially strong signal in the junctional zone. The fetal-derived junctional zone of the mouse placenta contains two types of trophoblast cells: spongiotrophoblasts and glycogen cells [70]. The specific roles of these trophoblasts are unknown, though there are reports that the junctional zone possesses endocrine functions as it has been found to release prolactin-like hormones [71]. There was also clustered iNOS antibody staining at the site of abscesses within the labyrinth of the placenta. The labyrinth is the site of gas and nutrient exchange between maternal and fetal blood and is the equivalent of the chorionic villi of human placenta. The labyrinth is composed of syncytiotrophoblasts, endothelial cells, and trophoblast giant cells, and it also contains fetal-derived macrophages [72]. Placental macrophages are significant mediators of gestational innate immunity, and are likely participants in NO production and induction of NO in surrounding tissues [73,74,75,76,77,78]. 

We followed WT and iNOS- mice with GBS chorioamnionitis over time to assess the role of NO on pregnancy and neonatal outcomes. WT mice developed GBS chorioamnionitis following intravaginal GBS inoculation less than iNOS- mice, although this difference did not reach significance. However, WT mice that developed GBS chorioamnionitis delivered preterm, 48–72 h after inoculation, more frequently than iNOS- mice. This suggests that WT mice are less susceptible to developing GBS chorioamnionitis, but once GBS establishes infection of the chorion and amnion with dissemination to the placenta, NO leads to detrimental inflammatory response and host tissue damage that contributes to preterm delivery. These differences do not seem to be driven by IUFD, for which rates between WT and iNOS- mice were roughly the same.

The murine model of chorioamnionitis used in this study reflects infection that has occurred by vertical ascension of GBS following vaginal colonization, which is the most common pathway that occurs in humans. Chorioamnionitis typically follows dissemination of bacteria from the genitourinary tract of mothers to the fetal membranes, amniotic fluid, and placenta [79]. This is supported by the fact that bacteria recovered from fetal membranes and amniotic fluid are often the same as those seen in the colonizing genitourinary flora such as *Ureaplasma*, anaerobes, GBS, and *E. coli* [80]. Less commonly, chorioamnionitis can develop by hematogenous spread from the mother to the placenta and directly into amniotic fluid following invasive intrauterine procedures. Therefore, the murine model of chorioamnionitis and effects on pregnancy and neonatal outcomes in this study reflects the same pathogenesis of chorioamnionitis as humans.

We sought to identify the molecular mechanisms that GBS utilizes to survive in a NO-rich environment. All 10 GBS capsular serotypes were evaluated for growth patterns in this study as it was unknown what response, if any, GBS would display to this specific form of oxidant stress and, secondly, to see if the serotypes that commonly cause disease were more resistant. First, we found that all 10 GBS capsular serotypes had specific patterns of growth delay when exposed to DETA NONOate, a NO adduct. GBS capsular serotype Ia did not start to grow at the lowest concentrations of DETA NONOate until approximately 16.5 h, whereas GBS strain CNCTC 10/84, capsular serotype V, was the only GBS strain to grow at 10 mM DETA NONOate. In contrast, none of the GBS capsular serotypes grew at 20 mM DETA NONOate, the highest concentration. These differences in growth delay showed that GBS can withstand NO exposure though there is a threshold that impairs survival, and that tolerance varies by strain. While the capsule and its components were not thought to have directly contributed to growth differences, the capsule types represent the diversity of GBS and highlight that there may be additional strain-specific genetic adaptations that allow for NO resistance.

Among the strains we tested, CNCTC 10/84 was best able to tolerate the microbicidal effects of NO. GBS strain CNCTC 10/84 was originally isolated from a septic neonate and demonstrates increased virulence in animal models [37,65,81,82]. This increased virulence is secondary to a hyperhemolytic phenotype due to increased production of β-hemolysin/cytolysin, a pigmented cytotoxin. βH/C increases GBS survival within macrophages [42,58]. This led us to suspect that βH/C was contributing to GBS strain CNCTC 10/84′s tolerance of higher NO concentrations. In the presence of DETA NONOate, GBS strain CNCTC 10/84 was able to hemolyze human adult red blood cells, though the efficiency decreased as DETA NONOate concentration increased. Comparing growth and survival of wild type CNCTC 10/84 with 10/84 Δ*cylE* in a NO-rich environment revealed that the non-hemolytic 10/84 Δ*cylE* strain had both significantly decreased growth and decreased survival in DETA NONOate. This indicated that βH/C is a contributor to GBS tolerance of NO stress and suggests one way that GBS is able to persist in the placenta despite increased NO production during chorioamnionitis.

The range of tolerance to DETA NONOate among the GBS strains we tested suggested genetically encoded stress responses, which we explored with RNA-seq of CNCTC 10/84 and a serotype Ia comparator, GBS strain A909, in the presence or absence of NO. Both GBS strains demonstrated numerous differentially expressed genes. The most upregulated gene shared by both GBS strains was a bifunctional acetaldehyde-CoA/alcohol dehydrogenase. Bifunctional acetaldehyde-alcohol dehydrogenase is an enzyme that converts acetyl-CoA to ethanol under anaerobic conditions [83]. Although not directly involved in scavenging reactive nitrogen byproducts, activation of anaerobic metabolism during NO exposure may represent an adaptive response that decreases the total burden of harmful radicals by limiting endogenous production of reactive oxygen species [84,85,86]. Bifunctional acetaldehyde-alcohol dehydrogenase has also been associated with increased bacterial adherence in *Listeria monocytogenes* and *Streptococcus pneumoniae* [87,88]. 

We wondered whether *cyl* genes would be significantly upregulated in response to NO exposure, leading to increased βH/C production. However, that was not the case. The CovS/CovR (control of virulence) two-component system responsible for transcriptional regulation of the *cyl* operon [89] did not show a significant response either. The *cyl* operon that encodes βH/C biosynthetic enzymes is downregulated by the CovR regulator, which is minimally expressed in CNCTC 10/84 due to a mutation in the *covR/covS* promoter [65]. This renders *cyl* operon expression constitutive in hyperhemolytic CNCTC 10/84. The CovR/CovS system is believed to function normally in A909, however. Together our results suggest that neither expression of the CovR/CovS system nor the *cyl* operon are directly mediated by NO stress.

The significantly downregulated genes largely fell under the class of ATP-binding cassette transporters, with the YBBM-related iron export transporter (*fetB*) gene having the most decreased expression shared between the two tested strains. This gene may be downregulated to increase iron availability for detoxification of NO. One such reaction that has been seen in *E. coli* and *B. subtilis* occurs via flavohemoglobins that are able to bind NO, which reacts with heme-bound iron [90,91]. These RNA-seq results provide a starting point for exploring and validating the other genes that are involved in GBS tolerance of NO. 

There are no other studies, to the best of our knowledge, that have evaluated the role of NO in GBS chorioamnionitis. The relationship of βH/C and NO production has been studied in the murine RAW 264.7 cell line, in which a hyperhemolytic variant of GBS strain COH1 (capsular serotype III) induced iNOS expression and NO production [92]. This provides insight into the potential cell types that are expressing iNOS in our murine model of GBS chorioamnionitis, as macrophages are present in the labyrinth of the placenta, which was one site of iNOS localization. Future work will evaluate the transcriptomic responses of other commonly isolated GBS capsular serotypes in neonatal infection, such as capsular serotype III [93]. 

In conclusion, this study utilized a two-pronged approach of evaluating the host NOS response to GBS invasion of the placenta and the response of GBS to host NO production during chorioamnionitis. We established that the murine placenta increases expression of iNOS in response to GBS chorioamnionitis, particularly in the glycogen cells of the junctional zone. Future work will identify the specific cell types that are expressing NOS and producing NO in the placenta. On the bacterial side, our work supports the hypothesis that βH/C contributes to increased virulence through improved tolerance to oxidant stress. Additional transcriptional data provides insight into the other pathways that may be involved in withstanding the effects of NO. These findings help define host NO-mediated responses to GBS chorioamnionitis and bacterial defenses that are activated in response. These results could ultimately inform pharmaceutical approaches, such as an iNOS inhibitor or antioxidant, to decrease the rates of preterm labor and sequelae of preterm birth on neonates in the setting of GBS chorioamnionitis. 

## Figures and Tables

**Figure 1 pathogens-11-01115-f001:**
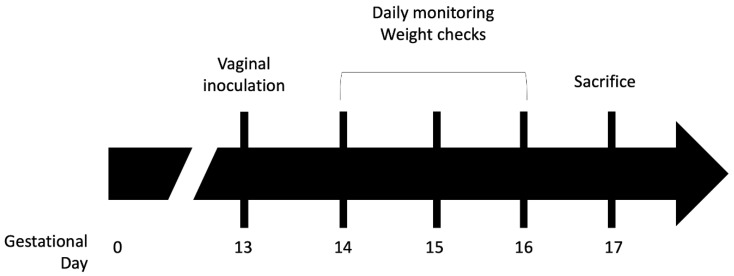
**Murine model of GBS chorioamnionitis.** Pregnant wild type, C57BL/6J, and iNOS-, B6.129P2-*NOS2^tm1Lau^*/J, dams were intravaginally inoculated with GBS or sham under anesthesia on day 13 of gestation. The dams were monitored daily for general wellbeing, weight gain or loss, and preterm delivery. Mice that did not deliver by gestation day 17 were sacrificed, and the placentas and fetuses were collected for RNA extraction, snap freezing, or paraformaldehyde fixation. Adapted from [41].

**Figure 2 pathogens-11-01115-f002:**
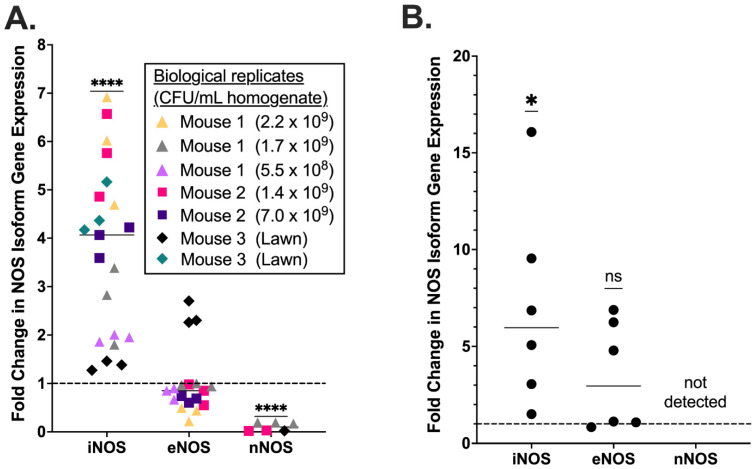
**Placental NOS expression following GBS infection.** RT-qPCR was performed on RNA isolated from murine placentas following chorioamnionitis with GBS strain CNCTC 10/84. Inducible NOS (iNOS) expression was found to be significantly increased, whereas endothelial NOS (eNOS) was unchanged and neuronal NOS (nNOS) was downregulated. Each shape represents an individual mouse and each color represents an individual placenta tested in triplicate. The inset table shows GBS CFU density for the mouse placentas analyzed. Lawn indicates full coverage of plate with bacteria that is unable to be quantified (**A**). Human chorionic villi dissected from two freshly collected placentas delivered by cesarean section in the absence of labor or other infectious or inflammatory complications was infected ex vivo with GBS strain CNCTC 10/84 then used for RNA isolation and RT-qPCR in triplicate technical replicates to assess expression of NOS isoforms (**B**). iNOS showed significantly increased expression following GBS exposure, compared to sham infected controls. eNOS expression was not significantly affected. nNOS was not detected from infected or control samples. (**** *p* < 0.0001, * *p* < 0.05; one sample t and Wilcoxon test; dotted lines indicate expression in the non-infected normalization control samples; horizontal lines within data points indicate median values).

**Figure 3 pathogens-11-01115-f003:**
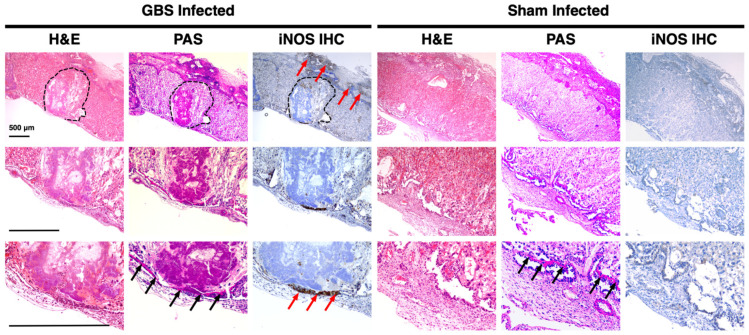
**Histopathology and immunohistochemical staining of murine placenta for iNOS localization.** Placentas from dams with GBS chorioamnionitis developed areas of abscess (outlined in dotted line) extending from the junctional zone into the labyrinth. Periodic acid-Schiff (PAS) staining revealed a band of dense pink staining in the glycogen cells of the junctional zone (black arrows). Immunohistochemical (IHC) staining for iNOS expression revealed diffuse upregulation with strong localized expression in the junctional zone and areas of abscess (red arrows). Placentas from sham-inoculated dams displayed less iNOS antibody staining and retained normal cellular and anatomic structural organization. (H&E: hematoxylin and eosin).

**Figure 4 pathogens-11-01115-f004:**
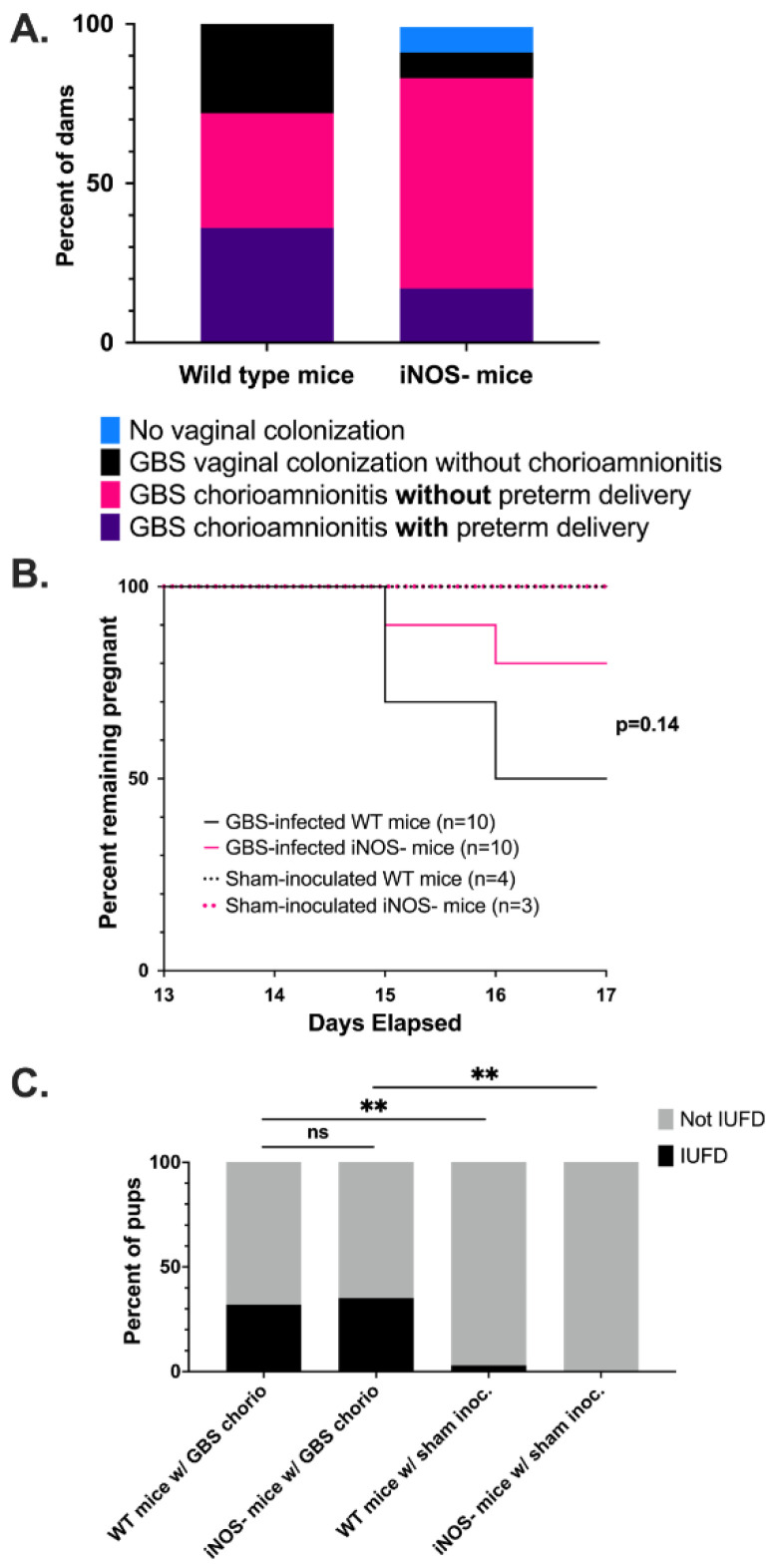
**Maternal, pregnancy, and fetal outcomes of mice with GBS chorioamnionitis.** Pregnancy outcomes of WT and iNOS- mice following induced GBS chorioamnionitis or sham infection are illustrated ((**A**), n = 14 for WT, n = 12 for iNOS-). Among the 10 mice in each condition that developed microbiologically confirmed GBS chorioamnionitis, 2 iNOS- mice delivered prematurely compared to 5 WT (*p* = 0.14 by log-rank test). No sham-infected mice from either group delivered prematurely (**B**). Both WT and iNOS- mice with GBS chorioamnionitis had significantly more IUFD than sham-inoculated WT and iNOS- mice (** *p* < 0.01 by chi-square test). The percent of IUFDs between WT and iNOS- dams with GBS chorioamnionitis were similar (**C**).

**Figure 5 pathogens-11-01115-f005:**
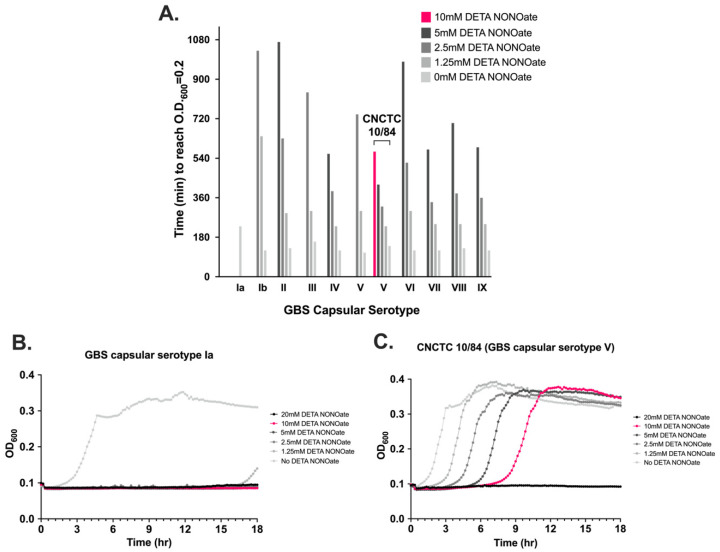
**Growth patterns of GBS during NO exposure.** All GBS capsular serotypes grew in the presence of NO though the degree of growth delay varied between serotypes (**A**). GBS CNCTC 10/84, a capsular serotype V, had better growth at all DETA NONOate concentrations compared to other strains such as our GBS capsular serotype Ia strain (**B**,**C**). Diethylenetriamine/nitric oxide adduct (DETA NONOate) releases nitric oxide into aqueous solution at a steady rate.

**Figure 6 pathogens-11-01115-f006:**
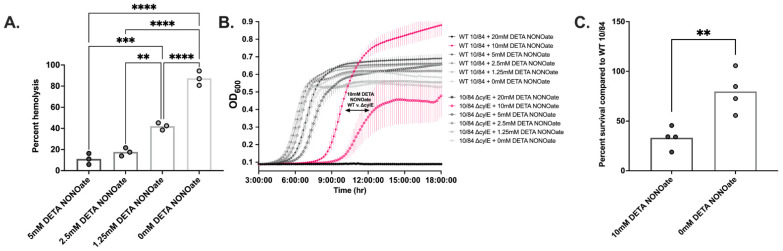
**β-hemolysin/cytolysin confers a survival advantage to GBS in a NO-rich environment.** βH/C-mediated hemolysis by GBS strain CNCTC 10/84 was present but impaired by DETA NONOATE ((**A**), ** *p* < 0.01, *** *p* < 0.001, **** *p* < 0.0001 by one-way ANOVA with Bonferroni’s multiple comparisons test). GBS strain 10/84 Δ*cylE*, which does not produce βH/C, had more growth delay compared to WT 10/84 under elevated NO conditions, best seen at 10mM DETA NONOATE in pink ((**B**), double arrow; average values are indicated for each time point with error bars indicating standard error across three technical replicates). GBS strain 10/84 Δ*cylE* also had decreased survival compared to WT 10/84 when exposed to NO (**C**). Experiments in this figure were conducted twice with 3–4 technical replicates each time. Results are shown from a single representative experiment. (** *p* < 0.01 by unpaired t test; columns extend to median values).

**Figure 7 pathogens-11-01115-f007:**
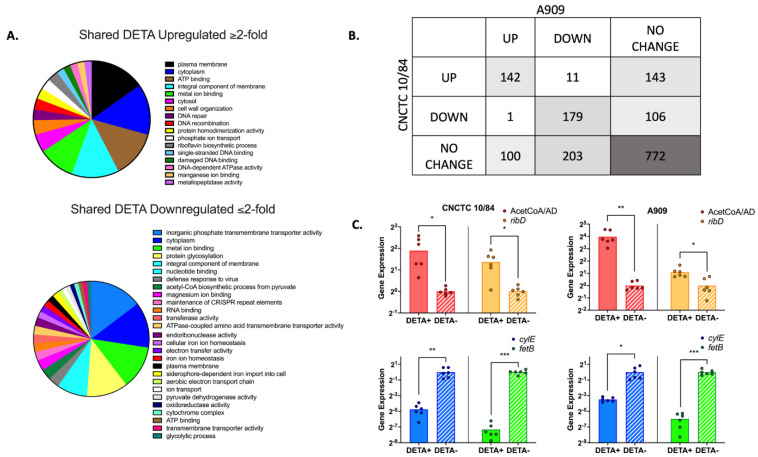
**RNA-seq of two GBS strains indicates extensive, shared transcriptome alteration in response to NO exposure.** Significantly upregulated and downregulated gene COG functional classes shared between the two GBS strains tested (**A**). Comparison of ortholog expression in the two GBS strains tested showed overlapping transcriptomic profiles (**B**). RT-qPCR validation of four genes with DETA-responsive expression differences by RNA-seq. Each RT-qPCR assay was run on two biological replicates, each with triplicate technical replicates (**C**). (* *p* < 0.05, ** *p* < 0.01, *** *p* < 0.005 by ANOVA with Dunnett’s correction for multiple comparisons).

**Table 1 pathogens-11-01115-t001:** Orthologous genes ≥4-fold upregulated by NO exposure in A909 and CNCTC 10/84.

A909 (Serotype Ia) Gene Locus	10/84 (Serotype V) Gene Locus	PANTHER Consensus Gene Annotation	A909 LFC	10/84 LFC	Mean LFC
SAK_RS00425	W903_RS00440	alcohol dehydrogenase	4.43	3.32	3.87
SAK_RS00805	W903_RS00840	disulfide oxidoreductase	3.56	2.84	3.20
SAK_RS10680	W903_RS10090	hypothetical protein	3.48	2.60	3.04
SAK_RS03085	W903_RS03140	hypothetical protein	3.58	2.28	2.93
SAK_RS10170	W903_RS09695	pyruvate formate-lyase-activating enzyme	2.75	3.02	2.89
SAK_RS10175	W903_RS09700	hypothetical protein	2.43	2.94	2.69
SAK_RS10190	W903_RS09715	anaerobic ribonucleoside-triphosphate reductase	2.31	2.88	2.60
SAK_RS10585	W903_RS09995	hypothetical protein	2.08	2.95	2.51
SAK_RS04360	W903_RS04040	cytosine deaminase	2.47	2.53	2.50
SAK_RS10180	W903_RS09705	oxidoreductases	2.28	2.65	2.46
SAK_RS04365	W903_RS04045	riboflavin synthase alpha chain	2.58	2.16	2.37
SAK_RS00720	W903_RS00750	GRPE protein	2.39	2.33	2.36
SAK_RS10400	W903_RS09920	response regulator of two-component system	2.36	2.21	2.29
SAK_RS04370	W903_RS04050	GTP cyclohydrolase II-related	2.53	2.03	2.28
SAK_RS10275	W903_RS09800	hypothetical protein	2.30	2.09	2.19
SAK_RS10645	W903_RS10055	hypothetical protein	2.08	2.27	2.18
SAK_RS10230	W903_RS09755	hypothetical protein	2.00	2.13	2.07

**Table 2 pathogens-11-01115-t002:** Orthologous genes ≤4-fold downregulated by NO exposure in A909 and CNCTC 10/84.

A909 (Serotype Ia) Gene Locus	10/84 (Serotype V) Gene Locus	PANTHER Consensus Gene Annotation	A909 LFC	10/84 LFC	Mean LFC
SAK_RS04090	W903_RS03770	inner membrane protein YBBM-related iron transporter (fetB)	−3.19	−4.23	−3.71
SAK_RS04085	W903_RS03765	hypothetical protein	−3.02	−4.18	−3.60
SAK_RS07535	W903_RS07235	membrane component of amino acid ABC transporter	−3.22	−3.47	−3.34
SAK_RS07540	W903_RS07240	hypothetical protein	−3.22	−3.30	−3.26
SAK_RS06865	W903_RS06570	5′-nucleotidase-related	−3.16	−3.34	−3.25
SAK_RS04495	W903_RS04200	hypothetical protein	−2.87	−2.25	−2.56
SAK_RS04500	W903_RS04205	histidine transport system permease protein (hisM)	−2.60	−2.37	−2.49
SAK_RS06330	W903_RS06145	hypothetical protein	−2.38	−2.27	−2.33
SAK_RS04505	W903_RS04210	D-methionine-binding lipoprotein (metQ)	−1.93	−2.59	−2.26
SAK_RS07995	W903_RS07675	hypothetical protein	−2.26	−2.20	−2.23
SAK_RS06205	W903_RS06010	s1 RNA-binding domain-containing protein 1	−2.26	−2.15	−2.21
SAK_RS06325	W903_RS06140	hypothetical protein	−2.11	−2.22	−2.16
SAK_RS08985	W903_RS08565	glutamine synthetase	−2.00	−2.04	−2.02

## Data Availability

Illumina sequencing reads from RNA-seq of 10/84 and A909 under NO exposure and control conditions are publicly available under the National Center for Biotechnology Information Sequence Read Archive BioProject accession PRJNA882489.

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
