# Peer review of "Nitric Oxide Production and Effects in Group B Streptococcus Chorioamnionitis"

_pathogens, 2022, doi:10.3390/pathogens11101115_

Round 1
Reviewer 1 Report
Keith et al has done a nice piece of work and I understand this is one of the good research projects to understand the pathophysiology and host-pathogen interaction of GBS in chorioamnionitis. However, there are few minor corrections to be done.
1. Please explain the existing literature on proposed method of GBS chorioamnionitis initiation and how the current research findings match/differ from it.
2. GBS serotype III subtypes have not been tested on this experiment. Please make a note on that as a limitation.
Author Response
- Please explain the existing literature on proposed method of GBS chorioamnionitis initiation and how the current research findings match/differ from it.
The proposed modes of transmission of chorioamnionitis described in the literature have been added and compared to our murine model of chorioamnionitis in the discussion of the manuscript.
- GBS serotype III subtypes have not been tested on this experiment. Please make a note on that as a limitation.
A statement highlighting the fact that GBS serotype III was not tested in these experiments has been added to the discussion of the manuscript.
Reviewer 2 Report
This is an interesting study investigating how NO production contributes to GBS disease, and explores the genetic basis of NO resistance using RNA-Seq. Overall, the manuscript is well written and presented logically. Some comments are as follows:
· Figure 6A – is there a no bacteria control for the hemolysis assays? Does the increasing amounts of DETA NONOate affect hemolysis of the RBCs?
· The authors use 10 different serotypes of GBS to investigate how they perform under NO exposure. What is the relevance of capsule type in these assays, seeing how 10/84 grows much better than its other serotype V counterpart? It might be helpful if the authors emphasize that the different capsular serotypes were used as a representation of the diversity of GBS, and even then, there might be other genetic bases for NO tolerance/resistance among individual strains (ie covR/S for 10/84)
· Following up on the previous comment, it would be interesting to see how hemolytic the other 10 strains are, in comparison to 10/84. Would the Ia strain be the least hemolytic, and the type IV and VII strains be almost as hemolytic as 10/84? This would further add evidence that B-hemolysin contributes to GBS survival in NO-rich environments.
· The authors performed a series of experiments that show the contribution of cylE to resisting NO stress. However, their RNA-Seq data suggests otherwise, especially for 10/84. Could this be because cylE is constitutively expressed at a very high level in 10/84 anyway due to the mutations in covR/S, which may have affected the RNA-Seq results? I think it would be good if the authors could further explore/discuss the disconnect between the cylE mutant data and the RNA-seq data.
· The authors state that expression of the cyl operon is slightly downregulated in A909, and have a value of LFC 0.91. Do they mean -0.91? A LFC of 0.91 is about twice as expressed (upregulated).
· Typographical error – please go through the manuscript and italicise cylE for the mutant strain
Author Response
- Figure 6A – is there a no bacteria control for the hemolysis assays? Does the increasing amounts of DETA NONOate affect hemolysis of the RBCs?
The RBCs used in the assays were not directly exposed to DETA NONOate, so the hemolysis was not directly affected by oxidant stress. The GBS that were exposed to DETA NONOate were spun and washed with PBS prior to performing the hemolysis assays, thereby removing remaining DETA NONOate in solution. This question has been addressed and clarified in the “Hemolysis assays” section of methods in the manuscript.
- What is the relevance of capsule type in these assays, seeing how 10/84 grows much better than its other serotype V counterpart?
An explanation on the relevance of testing the response of all GBS capsular serotypes to NO in this study have been added to the discussion.
- Following up on the previous comment, it would be interesting to see how hemolytic the other 10 strains are, in comparison to 10/84. Would the Ia strain be the least hemolytic, and the type IV and VII strains be almost as hemolytic as 10/84? This would further add evidence that B-hemolysin contributes to GBS survival in NO-rich environments.
We performed these hemolysis assays and did not find a clear relationship between capsular serotype, growth in DETA NONOate, and hemolysis. Although there were differences in hemolysis among the various strains, that variable alone did not appear to explain resistance to NO stress.
- Further explore/discuss the disconnect between the cylEmutant data and the RNA-seq data.
Additional discussion and potential explanations for the disconnect between the cylE mutant data and RNA-seq data have been added to the manuscript.
- The authors state that expression of the cyl operon is slightly downregulated in A909, and have a value of LFC 0.91. Do they mean -0.91? A LFC of 0.91 is about twice as expressed (upregulated).
The RNAseq data showed that the cyl operon expression was downregulated with a LFC of -0.91 for A909 exposed to DETA NONOate. This sentence has been fixed to reflect the negative LFC for the cyloperon in A909.
- Typographical error – please go through the manuscript and italicise cylE for the mutant strain
All instances of cylE in the manuscript have been italicized to fix this typographical error.
Reviewer 3 Report
This paper is an excellent investigation on the role of nitric oxide (NO) in chorioamnionitis caused by GBS. The title should be revised to describe the gist of the results of the paper and exact main effect/role of nitric oxide. In the study design/methods portion, you can add a diagram to clearly show the aims of the study. The discussion should contain relevant published data that is confirmed/refuted by this paper's findings. In your aims you mentioned the role on NO production in pregnancy outcomes, but you did not relate this to GBS bacterial load or genetic characteristics. These two factors may have something to do with pregnancy/fetal/maternal outcomes. Some figures need improvement: (1) Figure 3, instead of just IHC, just label it with iNOS IHC; (2) Figure 7, this is too crowded; remove some and move to supplemental, or split into two or three images. In your conclusion, add some therapeutic/clinical importance of your findings.
Author Response
- The title should be revised to describe the results of the paper and exact main effect/role of nitric oxide.
While the title, “Nitric Oxide Production and Effects in Group B Streptococcus Chorioamnionitis” is quite broad in scope, this was chosen because our results are extensive, as they cover both the murine and human host-response to chorioamnionitis as well as bacterial-response to nitric oxide, and would require too lengthy of a title to address all findings of this paper.
- In the study design/methods portion, you can add a diagram to clearly show the aims of the study.
Thank you. We have added a graphical abstract to diagram the goals and major findings of the study.
- The discussion should contain relevant published data that is confirmed/refuted by this paper’s findings.
There are no other studies that have looked at the role of nitric oxide in cases of GBS-specific chorioamnionitis. All efforts were taken to compare our findings to available studies evaluating NO, chorioamnionitis by any bacterial cause, and/or b-hemolysin/cytolysin.
- Expand on how GBS bacterial load or genetic characteristics may play a role on NO production and pregnancy/fetal/maternal outcomes.
Additional discussion of GBS bacterial load and its role in NO production has been added to the manuscript in order to emphasize all results of this study.
- Figure 3—Label IHC as iNOS IHC
The labels in Figure 3 have been changed from “IHC” to “iNOS IHC” to directly specify the IHC target.
- Figure 7—This figure is crowded. Remove some and move to supplemental or separate into two or three images.
The contents of Figure 7 have been adjusted so that there are fewer panels to make the figure less crowded and easier to view by readers. The images that are no longer in Figure 7 have been moved to the supplemental materials.
- In the conclusion, add some therapeutic/clinical importance of your findings.
A statement about the clinical importance and translatability of our study has been added to the discussion.
Round 2
Reviewer 2 Report
This reviewer is satisfied with the revision.